# Can Shear Wave Elastography Help Differentiate Acute Tonsillitis from Normal Tonsils in Pediatric Patients: A Prospective Preliminary Study

**DOI:** 10.3390/children10040704

**Published:** 2023-04-10

**Authors:** Bunyamin Ece, Sonay Aydin

**Affiliations:** 1Department of Radiology, Kastamonu University, Kastamonu 37150, Turkey; bunyaminece@hotmail.com; 2Department of Radiology, Erzincan University, Erzincan 24100, Turkey

**Keywords:** acute tonsillitis, shear wave elastography, stiffness, children, ultrasonography, pediatric

## Abstract

Shear wave elastography (SWE) is a non-invasive imaging technique used to quantify the elasticity/stiffness of any tissue. There are normative SWE studies on tonsils in healthy children in the literature. The purpose of this study is to analyze the palatine tonsils in children with acute tonsillitis using ultrasound and SWE. In this prospective study, pediatric patients aged 4–18 years diagnosed with acute tonsillitis and healthy children were included. Those with antibiotic use, chronic tonsillitis, adenoid hypertrophy, and having chronic disease, immunodeficiency, and autoimmune disease, or any rheumatological disease were excluded. The volume and elasticity of palatine tonsil were measured via ultrasound and SWE. The study included 81 (46 female, 35 male) acute tonsillitis patients, and 63 (38 female, 25 male) healthy children between the ages of 4 and 18. Elasticity (kPa) values of tonsils were found significantly higher in the tonsillitis group (SWE-R: 25.39 ± 4.64, SWE-L: 25.01 ± 4.17) compared to the normal group (SWE-R: 9.71 ± 2.37, SWE-L: 9.39 ± 2.19) (*p* < 0.001). In the tonsillitis group, a significant positive correlation was found between tonsil volume and elasticity (r: 0.774, *p*: 0.002). In conclusion, in pediatric patients with acute tonsillitis, higher kPa values were obtained with SWE in the palatine tonsils.

## 1. Introduction

Palatine tonsils are lymphoepithelial tissues that are part of the mucosal immune system, and in the lateral oropharyngeal wall, they lie within the tonsillar fossa, which is bounded anteriorly and posteriorly by the mucosal arches containing the palatoglossus and palatopharyngeus muscles, respectively [1]. The palatine tonsil contains 10–30 tubular branched crypts that extend through the entire thickness of the organ and expand the surface of the tonsils [2]. They are located in a strategic area where the respiratory and digestive tracts meet to provide continuous lymphoid stimulation. They perform a significant function in the protection against foreign infections because of their locations [3].

Palatine tonsils grow rapidly in the first years of life due to their immunological function. Although the exact growth mechanism is unknown, it is thought to occur when external antigen presentation triggers/catalyzes lymphoid hyperplasia and tonsil parenchyma enlargement. Tonsil size is most prominent in childhood and is directly linked to bacterial load and the amount of B and T lymphocytes. Later, depending on age, tonsillar involution can be seen [2].

Tonsillitis occurs if the proliferation of pathogens in the lymphoid tissue exceeds the protective power of activated lymphoid and immunoglobulin-producing cells [2]. Acute tonsillitis is an acute inflammatory condition that affects the tonsillar tissues of the oropharynx and is most common in school-aged children. It affects nearly all children at least once in their lives. Infection typically starts as a superficial infection and can only progress through peritonsillar cellulitis to the endpoint of a potentially life-threatening peritonsillar abscess [4]. Due to the possibility for airway obstruction, acute tonsillar enlargement is especially important [5]. It is critical to recognize and treat acute tonsillitis because of the complications that might emerge in untreated individuals.

The evaluation of tonsillitis for most patients involves a physical examination, risk classification using scoring systems, and consideration of fast antigen testing or throat culture. However, it is important to begin with a complete history and physical exam before conducting any evaluation. This data can then be used to calculate a Centor Score, which takes into account the presence of a fever, tonsillar enlargement, and/or exudates, tender cervical lymphadenopathy, and the absence of a cough. Each finding is worth one point, but the criterion has been updated to include an age adjustment: patients aged 3 to 15 years receive an extra point, while patients aged 45 and older have one point deducted from their score [6,7].

Patients with a score of 0 to 1 generally do not require further testing or antibiotics. For those with a score of 2 to 3 points, rapid strep testing and throat culture are possible options. However, clinicians should consider testing and prescribing empiric antibiotics for patients with scores of 4 or above. Throat culture can be used alone or in combination with fast antigen testing to screen for Group A Beta-hemolytic Streptococcus. It is worth noting that although the fast antigen test is specific (88% to 100%), it is not very sensitive (61% to 95%), which means false negatives are possible [8].

In cases of complex illnesses, such as patients with unstable vital signs, a toxic appearance, difficulty swallowing or tolerating oral intake, or trismus, a more thorough evaluation may be necessary. This may include neck imaging and laboratory testing, such as a complete blood count and basic metabolic panel to check renal function [9].

A lateral neck X-ray is a cheap and easily accessible diagnostic tool that holds clinical value for tonsil and neck soft tissue evaluation. It can be used as a first-line investigation to assess whether there is an increase in the width of the soft tissues in front of the vertebrae, and it may also detect the presence of gas or an air-fluid level [10,11]. Additionally, for more detailed evaluation, magnetic resonance imaging (MRI) and computed tomography (CT) are commonly used methods to radiologically evaluate pathologies related to the palatine tonsil. However, these imaging methods have some disadvantages, including high costs, the necessity for sedation in children, and exposure to ionizing radiation. Recently, ultrasound (US) has been increasingly used to diagnose peritonsillar infections and to evaluate tonsils’ morphological and volumetric changes [12,13,14].

Shear wave elastography (SWE) is a non-invasive imaging technique used to quantify the elasticity/stiffness of any tissue. SWE is a useful technique, particularly for children, due to its low operator dependence, ease of use, and reproducibility [15]. The literature includes SWE studies of different organs, such as the liver [16], testis [17], breast [18], thymus [19], parotid [20], kidney, and spleen [21] in children and adolescents. In the palatine tonsils, the literature includes studies showing normal values with SWE in healthy children and adolescents [14,22]. However, to our knowledge, there are no reports in the literature of a SWE study evaluating tissue elasticity/stiffness in cases of acute tonsillitis. In this respect, our study is the first to examine SWE values in patients with acute tonsillitis.

The purpose of this study is to use US and SWE to analyze the palatine tonsils in children with acute tonsillitis, examine the effect of tonsillar infection on tissue stiffness, and compare to the normal population.

## 2. Materials and Methods

Ethics committee approval was obtained from the hospital’s local ethics committee for this prospective study (ethics committee no: E-457297017-3902100.137). In addition, informed consent was obtained from the parents before the transcervical US and SWE examinations. Data for the study were collected between January 2021 and April 2022.

### 2.1. Patient Selection

Tonsillitis group: Pediatric patients between the ages of 4 and 18 who were diagnosed with acute tonsillitis and volunteered to participate in the study were included in the tonsillitis group. We selected patients with acute tonsillitis from our hospital’s pediatric outpatient clinic, family physician, or pediatric emergency department, and all patients were examined at the same state university hospital. There was no private service or a paid inspection. The diagnosis of acute tonsillitis was primarily based on the patient’s medical history and physical examination findings. During a physical exam, a clinician decided to diagnose acute tonsillitis by examining the throat for signs of inflammation, such as redness, swelling, and the presence of pus or white spots on the tonsils and enlarged lymph nodes.

Healthy control group: Healthy child participants without active disease and between the ages of 4 and 18 were included. Healthy individuals without any known disease or signs of infection or fever during the examination were included in the control group.

The exclusion criteria were determined as follows: Being younger than 4 years old or older than 18 years old, having a history of tonsillectomy, having started antibiotic treatment before or using antibiotics for an existing disease, having any chronic disease, having known adenoid/tonsillar hypertrophy, having peritonsillar abscess and collection, having immunodeficiency or cancer, and having an autoimmune disease or any rheumatological disease that may affect the tonsils. In addition, patients with recurrent acute tonsillitis and chronic tonsillitis characterized by persistent inflammation and infection of the tonsils were excluded from the study.

As a result, bilateral tonsils of 81 patients in the tonsillitis group and 63 patients in the healthy control group were evaluated.

The patients’ age, sex, BMI, height, and weight were all recorded in order to look for possible correlations between these parameters and tonsil dimensions, volumes, and SWE stiffness values (kPa).

### 2.2. Ultrasound and Shear Wave Elastography Assessment

The tonsil US examination was conducted with the patient lying down in the supine position, with their head tilted back slightly. The patient’s neck was in extension and their head was turned towards the side being examined. After applying a clear gel to the patient’s neck over the area where the tonsils are located, then ultrasound examination started using high-frequency linear-array transducers (L12-3, 3–12 MHz).

First the submandibular gland was discovered, then the palatine tonsil was visualized as a well-circumscribed hypoechoic structure just deep within the submandibular gland. Longitudinal and transverse plane imaging was performed according to the extension of the tonsil. During the ultrasound, the practitioner sometimes asked the patient to swallow to observe the movement of the tonsils. The diameters of both the right and left tonsils were measured in the anteroposterior, transverse, and longitudinal planes (Figure 1).

The volume of tonsils was automatically calculated using the US machine utilizing three simple measurements. Tonsil elastography measurements were determined automatically by the SWE feature of the machine (Figure 2). It was important to avoid applying pressure to the probe and to keep the practitioner’s hand stable throughout ultrasonographic imaging. Elasticity values were measured in kilopascals (kPa). Elasticity values of the lesions were measured three times using three different ROIs of 1 cm^2^ from the different areas by the same observer, and the average of these measurements was recorded as the final data. The US and SWE examinations were conducted by a radiologist with ten years of experience.

On the day of presentation to the outpatient clinic, patients underwent US and SWE examinations without having to wait for a separate appointment. After the patients were examined in the outpatient clinic, they were directed to the radiology department for imaging. The majority of patients sought medical attention within the first 2–3 days of symptom onset when their symptoms were most severe. Chronic cases and recurrent acute cases were excluded from the study with as much care as possible.

### 2.3. Statistical Analysis

Data analysis was performed using the SPSS software for social sciences (version 20) for Windows (IBM SPSS Inc., Chicago, IL, USA). To determine if the data had a normal distribution, the Kolmogorov–Smirnov test was utilized. Mean and standard deviation were used to present numerical variables with a normal distribution, while variables with a non-normal distribution were reported as medians with minimum and maximum values. Categorical variables were reported using numbers and percentages. Fisher’s chi-square test was used to compare the percentages of sexes between the normal and acute tonsillitis groups. Tonsil elasticity and volume parameters were compared between groups using the Mann–Whitney U and Student’s *t* tests, and compared based on gender using the Mann–Whitney U test. The Wilcoxon test was utilized to compare the parameters on the right and left sides. To explore potential relationships between tonsil elasticity and age, height, and weight values, Spearman correlation analysis was utilized. Logistic regression analysis was used to examine potential relationships between tonsil elasticity and sex. A two-tailed value of *p* < 0.05 was considered statistically significant.

## 3. Results

In the study, 144 (84 females, 60 males) participants aged 4–18 years were included. Of these, 81 (46 females, 35 males) were in the tonsillitis group and 63 (38 females, 25 males) were in the control group. The age and sex data of the tonsillitis group and the control group are shown in detail in Table 1. In terms of mean age and sex, there was no significant difference between the tonsillitis and control groups (*p* > 0.05) (Table 1).

The mean kPa values obtained from the right and left palatine tonsils by SWE and the mean volume values of the right and left palatine tonsils are shown in detail in Table 2. Accordingly, in the comparisons made separately for the right and left tonsils, kPa values were found to be significantly higher in the tonsillitis group compared to the control group (*p* < 0.001, for the right and left tonsils). The volume measurements obtained for the right and left tonsils were statistically significantly higher in the tonsillitis group compared to the control group (*p* = 0.026, *p* = 0.021 for the right and left tonsils, respectively) (Table 2).

In the correlation analyses performed separately in the tonsillitis group, control group and total study group, no significant statistical correlation was found between the elasticity (kPa) with sex, height, weight, body mass index (BMI), and tonsil volume (*p* > 0.05) (Table 3). In the tonsillitis group, a statistically significant positive correlation was found between tonsil volume and elasticity (Table 3).

## 4. Discussion

In our study, we aimed to investigate the changes in elasticity and volume in acute tonsillitis by comparing palatine tonsil volume and SWE measurements with normal values in acute tonsillitis, which is one of the common pathologies in the pediatric patient group. For this, we applied SWE for quantitative evaluation immediately after the gray scale examination. According to the results of our study, a significant increase was found in tissue stiffness and tissue volume in the palatine tonsil compared to the normal group in pediatric patients with acute tonsillitis.

Ultrasonography is not typically the primary imaging modality used for assessing tonsils. However, given the limitations of CT and MRI, ultrasound can be a useful supplementary tool. In addition to being simple, noninvasive, and affordable, ultrasound can also be performed at the patient’s bedside. Moreover, previous studies have shown that ultrasound is a helpful tool for evaluating tonsils and that volume measurements obtained through ultrasound are consistent with actual values [23,24,25]. As a matter of fact, Asimakopoulos et al. [26] found that US is a suitable objective method for determining tonsil volume in pediatric patients. There is an inherent existence of some form of signal-dependent noise in ultrasound imaging systems. Speckle noise is inherently present in ultrasound images. Its inherent presence occurs during the image acquisition phase. This noise can make it difficult to identify structures and can also degrade the image quality. Despeckling is an important preprocessing step in ultrasound imaging and can improve the accuracy of diagnosis and treatment planning. Despeckling techniques are used to reduce or eliminate this noise while preserving the underlying information. Much research has been completed in this field to remove speckle noise while preserving medical information in the image. The ultrasound devices which were used in the current study have this technology, so that we can acquire clear US and SWE images, and a successful sonographic imaging and study [27,28,29,30].

Shear wave elastography is a new method used to test tissue stiffness in a variety of tissues and organs, most notably the liver, breast, and thyroid [31,32]. SWE has been increasingly used in recent years to noninvasively identify normal and malignant cervical lymph nodes, usually by measuring lymph node stiffness [32,33]. Unlike other imaging techniques, it has been stated in the literature that SWE is noninvasive, inexpensive, convenient and reproducible, and that it can be performed in real time to monitor the stiffness of tissues [33].

In the literature, there are studies that evaluate the measurement of tonsil volume using ultrasound and evaluate normal kPa values of the palatine tonsils with elastography. In the first published study by Asumakopoulos et al. [26], they compared tonsil volume measured by ultrasound with the actual volume obtained from tonsillectomy in 52 tonsils from 26 children with adenotonsillar hypertrophy, aged 2–6 years. The study found that the mean tonsil volume measured by ultrasound was 3.6–3.9 mL, which was similar to the actual volume. In a study comparing obesity and hepatosteatosis with tonsillar volume, Ozturk et al. [34] found mean tonsillar volumes ranging between 3.18 mL and 4.45 mL in 97 children with a mean age of 13–14. In another study by Ozturk et al. [35], the mean tonsil volumes were found to be 2.03 mL in the chronic tonsillitis group and 5.36 mL in the obstructive sleep apnea syndrome (OSAS) group among 67 pediatric patients with a mean age of 10 years. Aydin et al. [36] conducted a normative study on 274 healthy children with a mean age of 7 years and found a mean tonsil volume of 1.5 mL. Mengi et al. [37] in their study of 85 patients with recurrent tonsillitis and OSAS, of which 50 were children with a mean age of 5.7 years, found the mean actual and ultrasound volumes in children as 3.5 mL and 3.67 mL, respectively. In our study, we found the mean tonsil volume as 2.90–2.97 mL in the acute tonsillitis group with a mean age of 10.9 years, and 1.69–1.71 mL in the healthy control group. When compared to studies in the literature that included adenotonsillar hypertrophy cases and OSAS patients, the tonsillar volume results of our study were lower. This may be because we excluded cases of tonsillar hypertrophy, recurrent tonsillitis, and chronic tonsillitis from our study. Tonsil volume measurements in normal healthy children in the literature and the measurements in our healthy control group were similar. The palatine tonsil volume in the tonsillitis group was found to be higher compared to the healthy control group in our study. These data demonstrate an increase in tonsil volume in cases of acute inflammation, suggesting that ultrasonography can detect this. As far as we are aware, there are two study [14,22] in the English literature that examine the normal elasticity values (kPa) of palatine tonsils using SWE, with similar results to ours.

Our study focused on the evaluation with SWE in patients with acute tonsillitis of the palatine tonsil. In our investigation, the mean elasticity values in the tonsillitis group were found to be statistically significantly higher when compared to the healthy control group. Our study provides first and preliminary findings regarding the SWE elasticity values of tonsillitis; hence, we cannot compare our findings to those of earlier research. However, we can compare it with the studies in the literature on several infective-inflammatory disorders. Accordingly, similar to our study, Qin Qin et al. [33] reported an increase in SWE values in acute bacterial cervical lymphadenitis. Lee JM et al. [38], in their study on Kikuchi patients, diagnosed with the acoustic radiation force impulse (ARFI) elastography method, found an increase in ARFI elastography values in Kikuchi patients and reactive hyperplasic lymph nodes compared to normal lymph nodes, similar to our study. Yoǧurtçuoǧlu et al. [39] found significantly higher mean elastography values than normal healthy kidneys in their ARFI study in patients with acute glomerulonephritis. According to our study and the findings in the literature, we conclude that there is an increase in the elasticity values of the tissues and tension and stiffening in the tissues in cases of infection-inflammation. This could be related to the fact that inflammatory cell infiltration and tissue liquefaction [40] in the palatine tonsil cause a significantly increased tension in the palatine tonsil in acute tonsillitis, resulting in high kPa values.

Studies in the literature have indicated that they evaluate shear wave flexibility in the hardest region of the lymph nodes due to the substantial regional variation in the distribution of lymphatic vessels in the SWE examination of the lymph nodes [41,42]. In our study, we performed measurements from solid areas while avoiding cryptic areas during SWE measurements in tonsils with increased volume. Similar to the literature, we think that tonsil flexibility can be evaluated more accurately in tonsils without heterogeneous internal structure.

Kamel et al. [43], in their study evaluating thyroid gland elasticity in pediatric patients with autoimmune thyroid disease, found thyroid SWE values to be significantly higher in children with autoimmune thyroiditis compared to healthy children due to parenchymal stiffness caused by lymphocytic infiltration and interstitial fibrosis. In the literature, a proportionally higher increase in tissue stiffness values has been found in autoimmune thyroiditis. However, when we look at our results, there is a moderate increase in proportion. One reason for this may be that fibrosis is not expected due to the pathophysiology of acute tonsillitis [5]. In addition, it is known that SWE may weaken while passing through the tissues due to its technical nature, and because the palatine tonsils are located relatively deeper than the thyroid tissue and lymph nodes, the push pulse may also weaken while passing through the tissues [44]. To compare the changes in the acute and chronic phases, there is a need for prospective longitudinal studies comparing acute tonsillitis and chronic tonsillitis and tonsillar hypertrophy patients in terms of tissue stiffness values.

Our study had some limitations. One of the limitations is the relatively small patient population. Therefore, the results may not be representative of the general population. Larger sample groups and multicenter studies are needed. Another limitation of our study is that patients with tonsillitis were not differentiated between viral or bacterial tonsillitis. Additional studies are needed to determine whether there will be a difference in SWE values in viral or bacterial tonsillitis. The major concern with US examinations is the high operator dependency variability, especially for quantitative SWE measurements. Another limitation is that the sonographic measurements were performed by only one physician. We opted to do so because extended examination times and reexamination were causing heightened discomfort and agitation among pediatric patients. Due to this situation, we were unable to assess the reproducibility of our measurements, and additional studies involving large patient groups and multiple practitioners are required to investigate the reproducibility of these measurements. Although this is a limitation of our study, the fact that all measurements were taken by a single physician ensured standardization between measurements in our study. Furthermore, although we determined cases with chronic tonsillitis and recurrent tonsillitis as exclusion criteria during patient selection, it was difficult to provide this strictly. It is challenging to assert that all patients are nonrecurrent and that this is the initial sickness. As a result, it is possible that we were unable to completely eliminate fibrotic changes during some tonsil evaluations.

## 5. Conclusions

The results of our study show that pediatric patients with acute tonsillitis exhibit higher kPa values in the palatine tonsils compared to the normal population, indicating increased stiffness. Additionally, we observed an increase in tonsil volume in acute tonsillitis. To the best of our knowledge, this is the first study to use SWE among patients with acute tonsillitis, and further prospective studies with larger populations are necessary to confirm these findings.

## Figures and Tables

**Figure 1 children-10-00704-f001:**
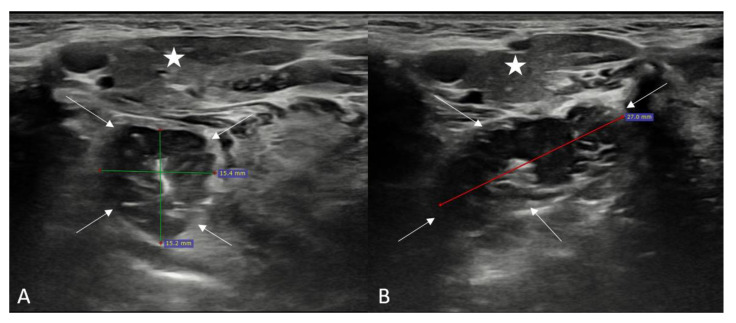
Ultrasonographic evaluation and measurements of the palatine tonsil in a healthy participant (Asterisk: Submandibular gland, Arrows: Palatine tonsil). (**A**): Transverse plane, (**B**): Longitudinal plane.

**Figure 2 children-10-00704-f002:**
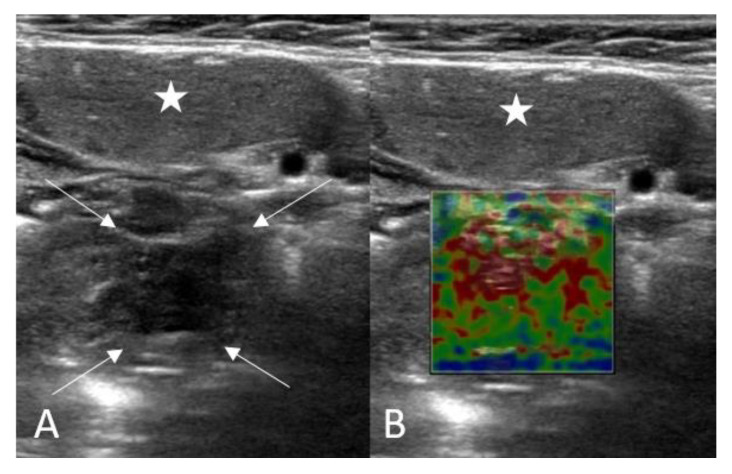
Shear wave elastography (SWE) evaluation of the palatine tonsil (**A**,**B**): Asterisk: Submandibular gland, Arrows: Palatine tonsil).

**Table 1 children-10-00704-t001:** Age and sex distribution of the participants.

	Tonsillitis Group (n = 81)	Control Group (n = 63)	*p* Value
Age * (year) (Mean ± SD)	10.9 ± 2.8	11.5 ± 3.2	0.442
Sex ** (Female/Male), n(%)	46(57)/35(43)	38(60)/25(40)	0.748

***** Student’s *t* test, ** Fisher’s Chi-square test, SD: Standard deviation.

**Table 2 children-10-00704-t002:** Data in the tonsillitis and healthy control groups, and comparison between groups.

	Tonsillitis Group (n = 81)	Control Group (n = 63)	*p* Value
SWE-R * (kPa) (Mean ± SD)	25.39 ± 4.64	9.71 ± 2.37	<0.001
SWE-L * (kPa) (Mean ± SD)	25.01 ± 4.17	9.39 ± 2.19	<0.001
Volume-R * (mm^3^) (Mean ± SD)	2.90 ± 0.63	1.71 ± 1.20	0.026
Volume-L * (mm^3^) (Mean ± SD)	2.97 ± 0.51	1.69 ± 0.80	0.021

***** Student’s *t* test, SWE: Shear Wave Elastography, kPa: Kilopascal, SD: Standard Deviation.

**Table 3 children-10-00704-t003:** Correlation analyses between elasticity and sex, weight, height, body mass index (BMI), tonsil volume (r = correlation coefficient; *p* = statistical significance).

	Tonsillitis Group (n = 81)		Control Group (n = 63)		Total Study Group (n = 144)	
	r	*p*	r	*p*	r	*p*
Sex/Elasticity *	0.452	0.163	0.313	0.396	0.423	0.260
Weight/Elasticity **	0.834	0.402	0.532	0.590	0.317	0.428
Height/Elasticity **	0.253	0.725	0.454	0.534	0.135	0.615
BMI/Elasticity **	0.345	0.243	0.318	0.712	0.192	0.533
Tonsil Volume/Elasticity **	0.774	0.002	0.160	0.663	0.065	0.618

* Logistic regression test, ** Spearman correlation test, BMI: Body Mass Index.

## Data Availability

The data that support the findings of this study are available from the corresponding author upon reasonable request.

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
