# Peer review of "Can Shear Wave Elastography Help Differentiate Acute Tonsillitis from Normal Tonsils in Pediatric Patients: A Prospective Preliminary Study"

_children, 2023, doi:10.3390/children10040704_

Round 1
Reviewer 1 Report
General comments:
Overall this is an interesting paper that makes a case for developing US, a non-invasive and relatively quick method, for evaluating tonsils. The discussion focuses on all of the other soft tissues where US is already used, with similar results, which makes a nice point of reference.
The methods used to diagnose acute tonsillitis in the "tonsillitis" group need to be explained.
There should be a more detailed description of how the tonsil ultrasound was performed. It is clear from the discussion that some user expertise is required, however the exact techniques and how the tonsil US were performed is required for others to be able to replicate this work.
The discussion provides valuable insights into how this study's evaluation of tonsils via US compares to different soft tissue assessments. It sounds like the big points are:
All evaluated tissues show a difference in elasticity based on inflammation, size measurement is accurate in a number of tissues, and there are differences in some tissue evaluations that may be more associated with fibrotic/chronic change, an area that was not explored in this study (it seems from the population selected that chronic disease was actually part of the exclusion criteria, this separation can be made more clear). The discussion would benefit from being organized to specifically make each point and back it up with information from the literature, directly comparing to the study results. This should not require significant modification.
Having a single operator test all of the patients should be mentioned in the methods as well as strengths/limitations. This is more of a limitation to the study than a strength; while it provides consistent testing it also limits the applicability of the findings since it is not clear that this is a replicable rather than technically challenging test. A comment on this in the discussion would be helpful. And while the numbers are not high, they were high enough to get a significant result and the stringency of limiting tonsillitis patients to those with acute, non-recurrent disease was a strength, since the parameters being measured were less likely to encompass multiple disease states and accidentally capture things like fibrotic change (as mentioned in the discussion).
Specific comments:
Line 50: often a lateral neck x ray is used to evaluate soft tissues in the head and neck, and although this obviously provides less detail than an MRI or CT it is 1. very common and 2. significantly less invasive/expensive than either. It would be helpful to comment on XR as well, just to ensure that all the different modalities are captured and US is compared appropriately.
Line 89 and 136, 137, 156, Table 1, Table 2: was the patient biological sex or gender identity recorded? It is important to use these terms correctly.
Line 135: Children as young as 4 are rarely described as Women/Men
Line 218: The discussion changes to size in this paragraph, then reverts to elasticity and stiffness in the following paragraph (line 223).
Author Response
Manuscript ID: children-2284333
Can Shear Wave Elastography Help Differentiate Acute Tonsillitis from Normal Tonsils in Pediatric Patients: A Prospective Preliminary Study
Dear Reviewer,
We appreciate the time and effort that you dedicated to providing feedback on our manuscript and are grateful for the insightful comments on and valuable improvements to our paper. We have incorporated most of the suggestions. Please see below for a point-by-point response to the comments and suggestions. And please see the attachment for the last edited version of the article with track changes enabled format-Word file.
1) General comments:
Overall this is an interesting paper that makes a case for developing US, a non-invasive and relatively quick method, for evaluating tonsils. The discussion focuses on all of the other soft tissues where US is already used, with similar results, which makes a nice point of reference.
The methods used to diagnose acute tonsillitis in the "tonsillitis" group need to be explained.
1.Response:
In line with your suggestions, we added the following sentences to the “material methods...patient selection...tonsillitis group” section.
"The diagnosis of acute tonsillitis was primarily based on the patient's medical history and physical examination findings. During a physical exam, a clinician decided to diagnose acute tonsillitis by examining the throat for signs of inflammation, such as redness, swelling, and the presence of pus or white spots on the tonsils as well as enlarged lymph nodes."
2) There should be a more detailed description of how the tonsil ultrasound was performed. It is clear from the discussion that some user expertise is required, however the exact techniques and how the tonsil US were performed is required for others to be able to replicate this work.
2.Response:
In line with your suggestions, we made a more detailed description of how the tonsil ultrasound were performed. The relevant part is below.
"The tonsil ultrasound examination was conducted with the patient lying down in the supine position, with their head tilted back slightly. The patient's neck was in extension and patient's head was turned towards the side being examined. After applying a clear gel to the patient's neck over the area where the tonsils are located, and then ultrasound examination started using high-frequency linear-array transducers (L12-3, 3-12 MHz).
First the submandibular gland was discovered, and then the palatine tonsil was visualized as a well-circumscribed hypoechoic structure just deep within the submandibular gland. Longitudinal and transverse plane imaging was performed according to the extension of the tonsil. During the ultrasound, the practitioner sometimes asked the patient to swallow to observe the movement of the tonsils. The diameters of both the right and left tonsils were measured in the anteroposterior, transverse, and longitudinal planes (Figure 1). "
3) The discussion provides valuable insights into how this study's evaluation of tonsils via US compares to different soft tissue assessments. It sounds like the big points are:
All evaluated tissues show a difference in elasticity based on inflammation, size measurement is accurate in a number of tissues, and there are differences in some tissue evaluations that may be more associated with fibrotic/chronic change, an area that was not explored in this study (it seems from the population selected that chronic disease was actually part of the exclusion criteria, this separation can be made more clear). The discussion would benefit from being organized to specifically make each point and back it up with information from the literature, directly comparing to the study results. This should not require significant modification.
3.Response:
In order to make it clearer that we excluded cases of chronic tonsillitis from the study, we separated it from the other exclusion criteria and added a separate sentence specifically excluding chronic tonsillitis.
“In addition, patients with recurrent acute tonsillitis and chronic tonsillitis characterized by persistent inflammation and infection of the tonsils were excluded from the study.”
In line with the suggestions of all reviewers, we made the discussion section more organized and detailed.
4) Having a single operator test all of the patients should be mentioned in the methods as well as strengths/limitations. This is more of a limitation to the study than a strength; while it provides consistent testing it also limits the applicability of the findings since it is not clear that this is a replicable rather than technically challenging test. A comment on this in the discussion would be helpful. And while the numbers are not high, they were high enough to get a significant result and the stringency of limiting tonsillitis patients to those with acute, non-recurrent disease was a strength, since the parameters being measured were less likely to encompass multiple disease states and accidentally capture things like fibrotic change (as mentioned in the discussion).
4.Response:
In line with your suggestions, we added in the limitations section that all patients were examined by a single practitioner. Additionally, as you suggested, we added the possibility that some patients may not have been clearly distinguished between acute and chronic tonsillitis during patient selection and that cases of chronic tonsillitis, despite being included in the exclusion criteria, may not have been entirely excluded from the study.
Expanded related sentences are below:
“The major concern with US examinations is the high operator dependency variability, especially for quantitative SWE measurements. Another limitation is that the sonographic measurements were performed by only one physician. We opted to do so because extended examination times and reexamination were causing heightened discomfort and agitation among pediatric patients. Due to this situation, we were unable to assess the reproducibility of our measurements, and further studies involving large patient groups and multiple practitioners are required to investigate the reproducibility of these measurements. Although this is a limitation of our study, the fact that all measurements were taken by a single physician ensured standardization between measurements in our study. Furthermore, although we determined cases with chronic tonsillitis and recurrent tonsillitis as exclusion criteria during patient selection, it was difficult to provide this strictly. It is difficult to say that all of the patients are nonrecurrent and that it is the first disease. As a result, it is possible that we were unable to completely eliminate fibrotic changes during some tonsil evaluations.”
5) Specific comments:
Line 50: often a lateral neck x ray is used to evaluate soft tissues in the head and neck, and although this obviously provides less detail than an MRI or CT it is 1. very common and 2. significantly less invasive/expensive than either. It would be helpful to comment on XR as well, just to ensure that all the different modalities are captured and US is compared appropriately.
5.Response:
In line with your suggestions, we have included sentences regarding X-ray in the section where imaging modalities are discussed. The updated version is as follows.
“A lateral neck X-ray is a cheap and easily accessible diagnostic tool that holds clinical value for tonsil and neck soft tissue evaluation. It can be used as a first-line investigation to assess whether there is an increase in the width of the soft tissues in front of the vertebrae, and it may also detect the presence of gas or an air-fluid level [6,7]. And for more detailed evaluation, magnetic resonance imaging (MRI) and computed tomography (CT) are commonly used methods to radiologically evaluate pathologies related to the palatine tonsil. These imaging methods, however, have some disadvantages, including high costs, the necessity for sedation in children, and exposure to ionizing radiation. Recently, ultrasound (US) has been increasingly used to diagnose peritonsillar infections and to evaluate the morphological and volumetric changes of the tonsils[8–10].
6) Line 89 and 136, 137, 156, Table 1, Table 2: was the patient biological sex or gender identity recorded? It is important to use these terms correctly.
6.Response:
We have recorded the biological sex.
Following your suggestions, we have updated the relevant sections to use 'sex' instead of 'gender', and 'male' and 'female' instead of 'men' and 'women'.
7) Line 135: Children as young as 4 are rarely described as Women/Men
7.Response:
Following your suggestions, we have updated the relevant sections to use 'sex' instead of 'gender', and 'male' and 'female' instead of 'men' and 'women'.
8) Line 218: The discussion changes to size in this paragraph, then reverts to elasticity and stiffness in the following paragraph (line 223).
8.Response:
Following your suggestions, we relocated the relevant paragraph to be the second paragraph of the discussion section.
Thank you very much for all your valuable comments, suggestions, contributions and efforts.

Reviewer 2 Report
Dear Authors
I found your work very interesting, original and clearly displayed.
I just have some considerations to do. The first one is that it is necessary to have a general English review from a native speaker because of some mistakes and lexical repetitions.
In the "introduction" section:
- - please give a precise definition of acute tonsillitis with diagnostic criteria, citing the literature in regard to these national/international criteria. Give information also if it is only a clinical diagnosis or if some laboratory tests or pharyngeal swab is needed.
- - what do you mean, clinically, for tonsillar hypertrophy? Give a definition of how paediatricians/surgeons define this entity, if there are some cut-off values and corresponding citations.
In the "materials and methods" section:
- - Tell more in detail how, where and in what context you selected patients with tonsillitis: in the emergency department? In others hospital departments? In a public or private context?
- - When and where did you perform the SWE examinations? At bedside? In the radiology service? In a private US studio? On what day was the US examination performed compared to the onset of symptoms?
- - As regard to colliquation or tonsillar abscess, did you find them? Did you include or exclude these patients from the study?
- - Write in this section (not in the discussion one) who performed the examinations, what qualification he/she has, and how many years of experience in US and SWE.
In the “discussion” chapter:
- - Insert a table with the correlation between the data about the tonsillar volume of normal children of your study and those reported in other significant studies, classifying it for ages if possible, and with the correlation between SWE tonsillar values in normal children of your study and those reported in other significant studies in literature.
- - Lines 208-210: add citations from literature that express the pathological process causing the changement in stiffness during inflammatory conditions.
- - Lines 215-217 are not very clear: what do you mean for cryptic areas and heterogeneous internal structure? Do non-solid areas can cause SWE artifacts?
Add another paragraph called “Limitations”.
There are too many repetitions in the “conclusions” paragraph: make a native English speaker correct it.
Author Response
Manuscript ID: children-2284333
Can Shear Wave Elastography Help Differentiate Acute Tonsillitis from Normal Tonsils in Pediatric Patients: A Prospective Preliminary Study
Dear Reviewer,
We appreciate the time and effort that you dedicated to providing feedback on our manuscript and are grateful for the insightful comments on and valuable improvements to our paper. We have incorporated most of the suggestions. Please see below for a point-by-point response to the comments and suggestions. And please see the attachment for the last edited version of the article with track changes enabled format-Word file.
1) Dear Authors
I found your work very interesting, original and clearly displayed.
I just have some considerations to do. The first one is that it is necessary to have a general English review from a native speaker because of some mistakes and lexical repetitions.
1.Response:
In line with your suggestions, we received professional language support for native english editing. Study has been edited and proofread by a native English speaker employed by Papercheck, LLC, 22nd day of March, 2023.
2) In the "introduction" section:
Please give a precise definition of acute tonsillitis with diagnostic criteria, citing the literature in regard to these national/international criteria. Give information also if it is only a clinical diagnosis or if some laboratory tests or pharyngeal swab is needed.
2.Response:
In line with your suggestions, we have included a new section in the introduction that discusses the diagnosis of acute tonsillitis. The following sentences have been added.
“The evaluation of tonsillitis for most patients involves a physical examination, risk classification using scoring systems, and consideration of fast antigen testing or throat culture. However, it's important to begin with a complete history and physical exam before conducting any evaluation. This data can then be used to calculate a Centor Score, which takes into account the presence of a fever, tonsillar enlargement and/or exudates, tender cervical lymphadenopathy, and the absence of a cough. Each finding is worth one point, but the criterion has been updated to include an age adjustment: patients aged 3 to 15 years receive an extra point, while patients aged 45 and older have one point deducted from their score[6,7].
Patients with a score of 0 to 1 generally don't require further testing or antibiotics. For those with a score of 2 to 3 points, rapid strep testing and throat culture are possible options. However, clinicians should consider testing and prescribing empiric antibiotics for patients with scores of 4 or above. Throat culture can be used alone or in combination with fast antigen testing to screen for Group A Beta-hemolytic Streptococcus. It's worth noting that although the fast antigen test is specific (88% to 100%), it's not very sensitive (61% to 95%), which means false negatives are possible[8].
In cases of complex illnesses, such as patients with unstable vital signs, a toxic appearance, difficulty swallowing or tolerating oral intake, or trismus, a more thorough evaluation may be necessary. This may include neck imaging and laboratory testing, such as a complete blood count and basic metabolic panel to check renal function[9].”
3) What do you mean, clinically, for tonsillar hypertrophy? Give a definition of how paediatricians/surgeons define this entity, if there are some cut-off values and corresponding citations.
3.Response:
We have detailed the introduction section in line with the recommendations of the reviewers.
4) In the "materials and methods" section:
Tell more in detail how, where and in what context you selected patients with tonsillitis: in the emergency department? In others hospital departments? In a public or private context?
4.Response:
We selected patients with acute tonsillitis from our hospital's pediatric outpatient clinic, family physician, or pediatric emergency department and all patients were examined at the same state university hospital. There was no private service or a paid inspection.
In line with your suggestion, we have provided additional details regarding this aspect in the patient selection criteria section. The updated version is as follows.
“Pediatric patients between the ages of 4 and 18 who were diagnosed with acute tonsillitis and volunteered to participate in the study were included to the tonsillitis group. We selected patients with acute tonsillitis from our hospital's pediatric outpatient clinic, family physician, or pediatric emergency department and all patients were examined at the same state university hospital. There was no private service or a paid inspection.”
5) When and where did you perform the SWE examinations? At bedside? In the radiology service? In a private US studio? On what day was the US examination performed compared to the onset of symptoms?
5.Response:
Following your suggestions, we have included these sentences in the Materials and Methods section.
“On the day of presentation to the outpatient clinic, patients underwent US and SWE examinations without having to wait for a separate appointment. After the patients were examined in the outpatient clinic, they were directed to the radiology department for imaging. The majority of patients sought medical attention within the first 2-3 days of symptom onset when their symptoms were most severe. Chronic cases and recurrent acute cases were excluded from the study with as much care as possible.”
6) As regard to colliquation or tonsillar abscess, did you find them? Did you include or exclude these patients from the study?
6.Response:
We encountered cases of collection and abscess, but we did not include them in the study.
Thanks to your critical reminder, we added cases with abscess and collection to the exclusion criteria.
7) Write in this section (not in the discussion one) who performed the examinations, what qualification he/she has, and how many years of experience in US and SWE.
7.Response:
We added the following sentence to the "Materials and Methods/Ultrasound and Shear Wave Elastography Assessment" section.
“The US and SWE examinations were conducted by a radiologist with ten years of experience.”
8) In the “discussion” chapter:
Insert a table with the correlation between the data about the tonsillar volume of normal children of your study and those reported in other significant studies, classifying it for ages if possible, and with the correlation between SWE tonsillar values in normal children of your study and those reported in other significant studies in literature.
8.Response:
In line with your suggestions, we have added a paragraph to the discussion section comparing our study's tonsil volumes results and SWE measurements with those reported in the literature.
“In the literature, there are studies that evaluate the measurement of tonsil volume using ultrasound and evaluate normal kPa values of the palatin tonsils with elastography. In the first published study by Asumakopoulos et al.[26], they compared tonsil volume measured by ultrasound with the actual volume obtained from tonsillectomy in 52 tonsils from 26 children with adenotonsillar hypertrophy, aged 2-6 years. The study found that the mean tonsil volume measured by ultrasound was 3.6-3.9 ml, which was similar to the actual volume. In a study comparing obesity and hepatosteatosis with tonsillar volume, Ozturk et al.[30] found mean tonsillar volumes ranging between 3.18 ml and 4.45 ml in 97 children with a mean age of 13-14. In another study by Ozturk et al.[31], the mean tonsil volumes were found to be 2.03 ml in the chronic tonsillitis group and 5.36 ml in the obstructive sleep apnea syndrome (OSAS) group among 67 pediatric patients with a mean age of 10 years. Aydin et al.[32] conducted a normative study on 274 healthy children with a mean age of 7 years and found a mean tonsil volume of 1.5 ml. Mengi et al.[33] in their study of 85 patients with recurrent tonsillitis and OSAS, of which 50 were children with a mean age of 5.7 years, found the mean actual and ultrasound volumes in children as 3.5 ml and 3.67 ml, respectively. In our study, we found the mean tonsil volume as 2.90-2.97 ml in the acute tonsillitis group with a mean age of 10.9 years, and 1.69-1.71 ml in the healthy control group. When compared to studies in the literature that included adenotonsillar hypertrophy cases and OSAS patients, the tonsillar volume results of our study were lower. This may be because we excluded cases of tonsillar hypertrophy, recurrent tonsillitis, and chronic tonsillitis from our study. Tonsil volume measurements in normal healthy children in the literature and the measurements in our healthy control group were similar. The palatine tonsil volume in the tonsillitis group was found to be higher compared to the healthy control group in our study. These data demonstrate an increase in tonsil volume in cases of acute inflammation, suggesting that ultrasonography can detect this. As far as we are aware, there are two study[14,22] in the English literature that examines the normal elasticity values(kPa) of palatine tonsils using SWE, with similar results to ours.”
9) Lines 208-210: add citations from literature that express the pathological process causing the changement in stiffness during inflammatory conditions.
9.Response:
In line with your suggestion, we have added a reference to the relevant sentence.
10) Lines 215-217 are not very clear: what do you mean for cryptic areas and heterogeneous internal structure? Do non-solid areas can cause SWE artifacts?
10.Response:
Thank you very much for your warning. We apologize for any confusion caused by the incorrect expression used due to a misspelling.
Based on your warning, we have made a correction in the following sentence: "Similar to the literature, we think that tonsil flexibility can be evaluated more accurately in tonsils with heterogeneous internal structure." we fixed our mistake as "without" instead of "with" in this sentence. We appreciate your help in improving the accuracy and clarity of our study.
11) Add another paragraph called “Limitations”.
11.Response:
We have added a sentence at the beginning of the last paragraph of the discussion section to summarize the limitations of our study. The new sentence is as follows: “Our study had some limitations.”
12) There are too many repetitions in the “conclusions” paragraph: make a native English speaker correct it.
12.Response:
We received professional language support for native english editing and they edited this section. The last version of conclusions as follows:
"The results of our study show that pediatric patients with acute tonsillitis exhibit higher kPa values in the palatine tonsils compared to the normal population, indicating increased stiffness. Additionally, we observed an increase in tonsil volume in acute tonsillitis. To the best of our knowledge, this is the first study to use SWE among patients with acute tonsillitis, and further prospective studies with larger populations are necessary to confirm these findings."
Thank you very much for all your valuable comments, suggestions, contributions and efforts.

Reviewer 3 Report
Based on preliminary examination of your submission, in our estimation it does not meet the standards, in terms of originality and novelty of the research. Therefore we must decline it without full peer review. Some of my major concerns for this rejection:
The content in the paper is very basic. organisation of paper is not correct. Some of the main concerns are not touched related to the topic.
Result section is very weak.
The complexity comparison should be made for validation of work which is highly missing.
Cross-Validation was lacking in this paper
Author Response
1)Based on preliminary examination of your submission, in our estimation it does not meet the standards, in terms of originality and novelty of the research. Therefore we must decline it without full peer review. Some of my major concerns for this rejection: The content in the paper is very basic. organisation of paper is not correct. Some of the main concerns are not touched related to the topic. Result section is very weak. The complexity comparison should be made for validation of work which is highly missing. Cross-Validation was lacking in this paper
1.Response:
We would like to express our sincere thanks to the reviewer for the valuable comments. We made significant changes in the material method and discussion section in response to reviewer and editor comments. We hope that the final version of the article will be of better quality and more satisfying. Please see the attachment for the last edited version of the article with track changes enabled format-Word file.

Round 2
Reviewer 3 Report
The literature of the work needs improvement. What will happen if the dataset provided is noisy in nature. The kind of noise that is found in ultrasound image is rayleigh noise, and since the ultrasound images are inherently noisy in nature. Then in that case how to handle such situation. Take reference from below papers and cite them in the revised manuscript.
Liu, F., Chen, L., Qin, P., Xu, S., Dong, Z., Zhao, X., ... & Qin, B. (2023). Is despeckling necessary for deep learning based ultrasound image segmentation?.
https://ieeexplore.ieee.org/abstract/document/10012299?casa_token=V6aU3PRyJo8AAAAA:1Zcwf16gHrcUGx9_zBxIfv5_i7BzwrIafLF-PSsuEkpBBbGTQ_w4zJjcU-mul2ASIpQ-647KTDiUGg
https://www.sciencedirect.com/science/article/pii/S0208521622000195
https://onlinelibrary.wiley.com/doi/abs/10.1002/ima.22851
Author Response
Manuscript ID: children-2284333
Can Shear Wave Elastography Help Differentiate Acute Tonsillitis from Normal Tonsils in Pediatric Patients: A Prospective Preliminary Study
Dear Reviewer,
We appreciate the time and effort that you dedicated to providing feedback on our manuscript and are grateful for the insightful comments on and valuable improvements to our paper. We have incorporated most of the suggestions. Please see below for a point-by-point response to the comments and suggestions. And please see the attachment for the last edited version of the article with track changes enabled format-Word file.
Reviewer 1 COMMENTS:
1) The literature of the work needs improvement. What will happen if the dataset provided is noisy in nature. The kind of noise that is found in ultrasound image is rayleigh noise, and since the ultrasound images are inherently noisy in nature. Then in that case how to handle such situation. Take reference from below papers and cite them in the revised manuscript.
Liu, F., Chen, L., Qin, P., Xu, S., Dong, Z., Zhao, X., ... & Qin, B. (2023). Is despeckling necessary for deep learning based ultrasound image segmentation?.
https://ieeexplore.ieee.org/abstract/document/10012299?casa_token=V6aU3PRyJo8AAAAA:1Zcwf16gHrcUGx9_zBxIfv5_i7BzwrIafLF-PSsuEkpBBbGTQ_w4zJjcU-mul2ASIpQ-647KTDiUGg
https://www.sciencedirect.com/science/article/pii/S0208521622000195
https://onlinelibrary.wiley.com/doi/abs/10.1002/ima.22851
1.Response:
In line with your suggestions, we read the articles you said and using the relevant references we added the following sentences to the “discussion” section.
“There is an inherent existence of some form of signal-dependent noise in ultrasound im-aging systems. Speckle noise is inherently present in ultrasound images. Its inherent presence occurs during the image acquisition phase. This noise can make it difficult to identify structures and can also degrade the image quality. Despeckling is an important preprocessing step in ultrasound imaging and can improve the accuracy of diagnosis and treatment planning. Despeckling techniques are used to reduce or eliminate this noise while preserving the underlying information. A lot of research has been done in this field to remove speckle noise while preserving medical information in the image. The ultra-sound devices which was used in the current study gladly has this technology, so that we can acquire clear US and SWE images, as well as a successful sonographic imaging and study[27-30].”
